Hydroxy proline and gamma-aminobutyric acid: markers of susceptibility to vine decline disease caused by the fungus Monosporascus cannonballus in melons (Cucumis melo L.)

Marquez Sixto Alberto 1 2 3
Crosby Kevin k-crosby@tamu.edu 1 2 3
Patil Bhimanagouda b-patil@tamu.edu 1 2 3 4
Avila Carlos 3 5
Ibrahim Amir MH 2 6
Pessoa Herika 7
Singh Jashbir 1 2 3
1 Department of Horticultural Sciences, Texas A&M University , College Station , TX , USA
2 Department of Horticultural Sciences, Texas A&M University, USDA National Center of Excellence for Melon at the Vegetable and Fruit Improvement Center , College Station , TX , USA
3 Department of Horticultural Sciences, Texas A&M University, Vegetable and Fruit Improvement Center , College Station , TX , USA
4 Department of Food Science and Technology, Texas A&M University , College Station , TX , USA
5 Department of Horticultural Sciences,Texas A&M Agrilife Research and Extension Center, Texas A&M University , Weslaco , TX , USA
6 Department of Soil and Crop Sciences, Texas A&M University , College Station , TX , USA
7 Agronomia, Universidad Federal de Viçosa-UFV , Viçosa , Minas de Gerais , Brasil
Singh Anshuman
Electronic publication date: 2023 Mar 2
Publication date: 2023
Volume: 11
Electronic Location ID: e14932
Received 2022 Nov 22; Accepted 2023 Jan 30
License: This is an open access article, free of all copyright, made available under the Creative Commons Public Domain Dedication. This work may be freely reproduced, distributed, transmitted, modified, built upon, or otherwise used by anyone for any lawful purpose.
License URL: https://creativecommons.org/publicdomain/zero/1.0/

Keywords: Vine Decline Disease, Monosporascus cannonballus, Hydroxy proline, Gamma-aminobutyric acid, Markers of susceptibility, Melons

Funding: USDA-NIFA-SCIR-2017-51181-26834 This research was funded by the USDA-NIFA-SCIR-2017-51181-26834 grant program. The funders had no role in study design, data collection and analysis, decision to publish, or preparation of the manuscript.

==============================
Background

Vine decline disease caused by the fungus Monosporascus cannonballus, is a threat to melon production (Cucumis melo L.) worldwide. Nonetheless, little is known about the metabolites produced during the host pathogen interaction. Thus, the objective of this study was to measure quantities of amino acids produced over time during such an interaction.

Methods

Two melon genotypes named TAM-Uvalde (susceptible) and USDA PI 124104 (resistant) were grown and inoculated with M. cannonballus. The metabolites previously stated were measured before inoculation (0 hours) and 24, 48 and 72 hours after inoculation, using high performance liquid chromatography analysis.

Results

The production of some amino acids during the interaction of the resistant and susceptible melon genotypes with the fungus M. cannonballus was different regarding quantities over time. Interestingly, hydroxy proline was always up-regulated in higher quantities in response to pathogen infection in the genotype TAM-Uvalde. Also, the up-regulation in higher quantities of gamma-aminobutyric acid in the genotype TAM-Uvalde 48 and 72 hours after inoculation, suggests more penetration of the pathogen in its roots. Hence, taken together, hydroxy proline and gamma-aminobutyric acid levels could be used as markers of susceptibility to vine decline disease caused by M. cannonballus, which could be useful in developing resistant varieties.

Introduction

The worldwide production of melons in 2019 was 28,467,920 tons and was worth more than one billion US dollars. Thus, it constitutes a valuable source of revenues for growers. Due to its popularity, melons are widely grown in many countries (FAO, 2020).

The fungus M. cannonballus is present in all the areas of the world where melons are grown, and this pathogen causes significant damage to melon production (Aleandri et al., 2017). Moreover, melon growers confront many problems, especially in Texas. Problems such as pests, lack of labor, competition for markets and diseases are factors that cause an impact on their production and their economic return. Vine decline disease caused by the pathogen M. cannonballus causes a significant damage to melon production because its control increases costs and pollutes the environment due to the use of chemicals. For instance, the fungus M. cannonballus has often been controlled with methyl bromide and other highly toxic fumigants (Crosby, 2001; Martin & Miller, 1996).

Martin & Miller (1996) indicate that M. cannonballus is a pyrenomycetes fungus, which is distinctive within the ascomycetes. It is homothallic and produces fertile perithecia in roots and releases one ascospore from one ascus. The sexual fruiting body is dark brown or black when reaching maturity and resembles a cannonball. Also, it contains between one to 16 nuclei per spore, but usually it has eight when mature. Its perithecia is spherical with a tiny neck and embedded in the root cortex. When it breaks, its spores are released into the soil. This fungus lacks conidial stage (Figs. 1, 2, 3, 4 and 5). The same authors reported that a rapid collapse of the vine takes place just before harvest because the absorption of water is blocked by the pathogen, which results in fruits with sunburn, low sugar content as well as a premature abscission from the pedicle before ripening and consequently, they become unmarketable. The symptoms previously described become more severe when the plant is under conditions that may generate stress. For example, heavy fruit load, drought, heat, and dense insect feeding.

Figure 1 Root with embedded perithecium.

Figure 2 Perithecium with spores.

Figure 3 Spores.

Figure 4 Plants in the field showing symptoms of vine decline disease caused by M. cannonballus.

Figure 5 M. cannonballus in V8 culture.

Amino acids are essential molecules found in all living organisms. Additionally, some amino acids are precursors of molecules that are involved in plant immunity such as ethylene (Burger & Chory, 2019). Furthermore, little is known about the compounds produced during the infection of melon plants with the pathogen M. cannonballus. Hence, ascertaining the amino acids produced during the infection of melon plants with the fungus M. cannonballus may be useful for formulating recommendations regarding the control of vine decline disease (Crosby, 2000).

It is hypothesized in this study that the amino acids produced during the infection of susceptible and resistant melon genotypes with a pathogen will be different regarding quantities over time. Therefore, the objective of this study was to measure quantities over time of amino acids produced during the infection of the susceptible TAM-Uvalde and the resistant USDA PI 124104 melon genotypes to vine decline disease with fungus M. cannonballus.

Materials & Methods

Location

An experiment was conducted between September and October 2021 at Texas A&M HortTrec facility, in College Station, Texas. Two genotypes were used for this study. A susceptible genotype, named TAM-Uvalde and a resistant genotype named USDA PI 124104. Previous studies based on visual assessments of root damage determined the susceptibility and resistance of the genotypes previously mentioned (Marquez et al., 2022). The plant material used in this experiment belonged to the melon breeding program of Texas A&M University, except for the genotype USDA PI 124104, which was obtained at the USDA North Central Regional Plant Introduction Station, located in Ames, Iowa and is originated in India. Additionally, it is important to remark that both genotypes can be distinguished by the morphology and color of their fruits, mainly (Fig. 6).

Figure 6 Fruits of the genotypes used in this study.

Left: USDA PI 124104 (resistant). Right: TAM-Uvalde (susceptible).

Inoculum production

The pathogen M. cannonballus, was isolated from infected roots following the procedure described by Marquez et al. (2022) and when spores were observed, a mixture of sand and ground oat hulls, combined at a rate of 45 g of oat hulls to 500 cm3 of sand was prepared. In 1 L flasks, 100 ml of water was combined with 500 cm3 of this medium and autoclaved twice for 60 min with a 1-day interval. The medium was inoculated with three, one cm2 pieces of fungi-colonized agar, cut from a V8 culture. The inoculated flasks were kept at room temperature under 12 h of fluorescent light/day for 5 weeks as previously described by Salari et al. (2013). Finally, the inoculum yielded 1.22 × 107 colony forming units of M. cannonballus per gram of sand medium.

Plant material and seedling development

An experimental design was used, which consisted of two genotypes (resistant vs susceptible) and two inoculation treatments (inoculated vs mock-inoculated control) for a total of four factorial treatments and three repetitions: RI, resistant (USDA PI 124104) inoculated with M. cannonballus, RC, resistant (USDA PI 124104) mock-inoculated, SI Susceptible (TAM-Uvalde) inoculated with M. cannonballus and SC susceptible (TAM-Uvalde) mock-inoculated.

Plants were grown under greenhouse conditions with an average temperature of 28 °C and 12 h of light period. Seeds of melons were germinated in trays with sterilized peat moss and 2-week-old seedlings were transplanted into trays of 38 cells, which hold a volume of 2,376 cm3 of medium (38 seedlings/tray).

Peat moss was used as a medium. It was sterilized in an autoclave for 30 min. Then, it was cooled down at room temperature for 24 h. Lastly, it was re-sterilized following the same procedure previously described. Each cell in a tray was filled halfway up with peat moss. Afterwards, 10 g of inoculum was added. Finally, more peat moss was added to each cell to fill it up completely.

Roots were sampled before inoculating the fungus, M. cannonballus, for the 0 h experiment. The rest of the plants were transplanted and taken out of the trays 24, 36 and 72 h after inoculating the fungus to sample their roots and build an amino acid profile. A total of 38 plants per treatment were taken out of the trays to sample their roots. Afterwards, the roots were washed to remove the medium. Lastly, they were bagged and kept frozen at −80 °C until proceeding to amino acid composition analysis.

Amino acids profiling

The frozen roots were ground in liquid nitrogen with a mortar and a pestle. 50 mg were transferred into 15 ml tubes. Afterwards, 3 ml of methanol were added to the tubes. The content was homogenized at 78.76 g for 2 min. Then, it was sonicated for 15 min and vortexed for 1 min. After vortexing the content, samples were centrifugated at 5008.64 g for 6 min at 10−15  °C. Finally, the supernatant was transferred into 15 ml tubes (filtrate I). The residue obtained during the first extraction, was mixed with 3 ml of methanol, and homogenized at 78.76 g for 2 min. Then, it was sonicated for 15 min and vortexed for 1 min. Afterwards, it was centrifuged at 5008.64 g for 6 min at 10−15 °C. Finally, the supernatant (filtrate II) was transferred into the previously labeled tube containing the filtrate I. The final volume was recorded, a derivation step was performed. A total of 700 µL were transferred into an amber color vial and then, 250 µL of dansyl chloride, 600 µL of sodium borate buffer and 100 µL of diamino heptane (internal standard) were added to it. Afterwards, it was vortexed for 1 min and incubated for 30 min in water bath at 60 °C. Following incubation, 60 µL of 2N acetic acid were added and samples were vortexed for 30 s to stop the reaction. Lastly, the content was transferred from the amber vials to the centrifuge tubes. The tubes were centrifuged at 7876 g rpm for 5 min and the clear supernatant that was obtained was transferred into vials for high performance liquid chromatography analysis (HPLC). A volume of 2 µl was injected into a HPLC system comprising a PerkinElmer Series 200 binary pump and autosampler (Shelton, CT, USA) for identification of amino acids and estimation of their relative abundance according by the procedure described (Singh et al., 2020). The optimization of HPLC analysis conditions and preparation of calibrations curves were carried out using authentic standards of amino acids procured from Sigma Aldrich. Also, the samples were kept frozen at −80 °C before and after analysis.

Statistical analysis

Statistical analyses were performed using statistical analysis system (SAS) PROC ANOVA (SAS Institute, 2020) and means were analized using least significant diference (LSD) 5%.

Results & Discussion

Analysis of amino acids

0 hours

The means of amino acids concentrations are displayed in Table 1. The concentration of glutamine, citrulline, serine, asparagine, glycine, beta-alanine, methionine, tyrosine, phenylalanine, isoleucine, leucine, hydroxy proline, valine, threonine, gamma-aminobutyric acid were significantly higher in the susceptible genotype TAM-Uvalde while the concentration of alanine was significantly higher in the resistant genotype USDA PI 124104.

Table 1 Amino acid quantities measured in samples of treatments RI, RC, SI and SC.

0 hours (pre-inoculation) and 24, 48 as well as 72 hours after inoculating plants with the fungus M. cannonballus.

		0 Hours	24 Hours	48 Hours	72 Hours	
Class of Metabolites	Compound	RI	RC	SI	SC	RI	RC	SI	SC	RI	RC	SI	SC	RI	RC	SI	SC	
Amino acids (µg/g fresh weight)	Arginine	n.d	n.d	n.d	n.d	n.d	n.d	n.d	n.d	n.d	n.d	n.d	n.d	n.d	n.d	n.d	n.d	
Glutamine	0 b	7.15 a	169.2 a	11.4 c	69.5 b	0 d	0.5 a	0 b	59.48 a	0 b	0 b	0 b	
Citrulline	0 b	0.44 a	9.59 a	0 b	0 b	0 b	45.22 a	0 b	45.22 a	0 b	0 b	0 b	77.91 a	0 b	
Serine	0 b	0.53 a	1.2 a	0 b	0 b	0 b	0 d	27.09 a	12.38 b	2.08 c	n.d	n.d	n.d	n.d	
Asparagine	42.83 b	362.25 a	661.01 a	260.4 b	72.4 c	89.90 b	290 a	0 d	n.d	n.d	n.d	n.d	
Threonine	27.66 b	70.19 a	58.62 a	11.02 b	58.62 a	11.02 b	23 a	0 b	0 b	0 b	31.08 a	3.15 b	2.05 c	0 d	
Glycine	0 a	0.16 a	1.75 a	0 b	0 b	0 b	697.1 a	161.2 b	0 c	0 c	0 b	0 b	32.85 a	0 b	
ß-Alanine	14.85 b	19.5 a	0 b	0 b	0 b	2.23 a	22.75 b	122.8 a	120.5 a	5.99 c	104.5 b	206.9 a	
Alanine	48.72 a	18.28 b	307.86 a	14.01 c	27.97 b	13.5 c	563 a	90.05 b	41.34 b	83.01 a	
γ-amino butyric acid	11.85b	19.5 a	26.23 a	4.85 b	26.23 a	4.65 b	0 c	0 c	189.2 a	4.72 b	59.29 c	60.48 c	91.52 a	84.19 b	
Methionine	11.97 b	27.97 a	8.53 b	29.67 a	0 b	11.82 a	0 b	11.82 a	18.67 d	32.04 c	54.04 a	38.81 b	
Tyrosine	0.08 b	4.52 a	n.d	n.d	n.d	n.d	n.d	n.d	n.d	n.d	0 b	0 b	0 b	0.38 a	
Phenylalanine	0 b	0.86 a	n.d	n.d	n.d	n.d	n.d	n.d	n.d	n.d	n.d	n.d	n.d	n.d	
Isoleucine	0 b	1.68 a	0 b	0 b	0 b	0.98 a	n.d	n.d	n.d	n.d	n.d	n.d	n.d	n.d	
Leucine	0 b	1.26 a	n.d	n.d	n.d	n.d	n.d	n.d	n.d	n.d	n.d	n.d	n.d	n.d	
Hydroxy Proline	0 b	0.11 a	0 b	0 b	4.76 a	0 b	0 b	0 b	33.64 a	0 b	0 b	0 b	42.66 a	0 b	
Valine	0 b	0.4 a	0 b	0 b	1.34 a	0 b	n.d	n.d	n.d	n.d	n.d	n.d	n.d	n.d	
Proline	n.d	n.d	n.d	n.d	n.d	n.d	n.d	n.d	n.d	n.d	n.d	n.d	n.d	n.d	n.d	n.d	
Histidine	n.d	n.d	n.d	n.d	n.d	n.d	n.d	n.d	n.d	n.d	n.d	n.d	n.d	n.d	n.d	n.d	
Notes.

significant differences (P < 0.05) among the treatments are shown by different letters, based on a post hoc least significant difference test (LSD). Different types of letters indicate means separation for the interaction genotype & inoculation, genotypes (Italics) and inoculation (bold).

RI resistant (USDA PI 124104) inoculated with M. cannonballus

RC resistant (USDA PI 124104) mock-inoculated

SI susceptible (TAM-Uvalde) inoculated with M. cannonballus

SC susceptible (TAM-Uvalde) mock-inoculated

n.d non-detected

24 hours

The means of the amino acid concentrations are displayed in Table 1. The amino acids glutamine, citrulline, serine, glycine, beta-alanine, alanine, hydroxy proline, isoleucine, and valine presented significant differences among treatments due to the interaction genotype × inoculation. The amino acids asparagine and methionine presented differences due to genotypes. Lastly, the amino acids threonine and gamma-aminobutyric acid presented significant differences between treatments due to inoculations.

48 hours

The means of the amino acid concentrations are displayed in Table 1. The amino acids serine, asparagine, threonine, glycine, beta-alanine, gamma-aminobutyric acid, and hydroxy proline presented significant differences among the treatments due to the interaction genotype × inoculation. The amino acids glutamine and alanine presented significant differences between the treatments due to genotypes. Finally, the amino acids citrulline and methionine presented significant differences between the treatments due to inoculations.

72 hours

The means of the amino acid concentrations are displayed in Table 1. The amino acids glutamine, citrulline, threonine, glycine, gamma-aminobutyric acid, methionine, tyrosine, and hydroxy proline presented differences among the treatments due to the interaction genotype × inoculation. Finally, the amino acids beta-alanine and alanine presented differences between the treatments due to genotypes.

It is commonly known that amino acids are part of all living organisms, and, in some cases, the concentration of amino acids was higher due to inoculations with the fungus, M. cannonballus. Thus, they may have been produced by the fungus. For example, Gong et al. (2007) indicated that asparagine, which is an essential amino acid derived from citrulline, is needed by fungus Coniothyrium minitans for conidiation. In other words, it needs it to reach its reproductive stage. Likewise, Canonica et al. (1979) reported that the fungus Cochliobolus miyabeanus requires methionine to produce metabolites derived from cochlioquinone. Therefore, it can be thought that the fungus M. cannonballus may have produced amino acids to carry out physiological functions. Also, the presence of the pathogen may have triggered the production of these amino acids in the plants regardless of the genotype used, which can be interpreted as a response of the plant that was reflected in significant differences due to the inoculation. Hence, further research, for instance, gene expression studies may be useful in determining their origin (Joshi et al., 2019). Similarly, the concentration of amino acids is linked to several factors. For instance, genotype, environment, and cultural practices (Bernillon et al., 2013). Therefore, the presence of the pathogen along with a defense response of the plant to counter the infection may have caused an increase in their concentration.

Citrulline is commonly present in plants belonging to the Cucurbitaceae family. Moreover, its presence is associated with responses to abiotic stresses. Interestingly, citrulline is an amino acid that is not translocated long distances within the plant and is found in higher concentrations within fruits of the same family (Joshi et al., 2019). In addition, citrulline is a precursor of arginine that is a precursor of nitric oxide, which has been reported to be produced by plants as a defensive compound (Vitor et al., 2013). However, it was not present in the roots (Table 1). Nevertheless, it is notable that citrulline was only present in the susceptible and resistant genotypes whose plants were inoculated (Table 1), which suggests that it is not involved in resistance to vine decline disease. However, citrulline may have been used as a signaling molecule. For example, molecules such as hydrogen sulfide and nitric oxide are important signaling molecules in plants. Therefore, citrulline may have been involved in the activation of a physiological process within the plant (Zang & Xie, 2021).

Gamma-aminobutyric acid is reported as a signaling molecule because it is produced rapidly when a plant is wounded or suffered from mechanical damage (Arçay et al., 2012; Busch & Fromm, 1999; Kinnersley & Turano, 2000). Remarkably, the concentration of this amino acid increased in both genotypes in the same proportion 24 h after inoculating the fungus, M. cannonballus (Table 1). However, the susceptible genotype TAM-Uvalde, exhibited higher quantities 48 and 72 h after inoculating the fungus M. cannonballus (Table 1), which suggests that the roots of the susceptible genotype suffered from more penetration of the fungus, M. cannonballus, than the roots of the resistant genotype USDA PI 124104.

Hypersensitive responses of plants have been related to the presence of calcium, which is involved in the production of GABA (Kinnersley & Turano, 2000). Xu & Heath (1998) reported that the concentration of calcium in cells of resistant cowpea plants rises to generate a hypersensitive response to counter the infection of cowpea rust fungus, Urumices vignae. Thus, the surge in the concentration of GABA previously described may have been an indication of a similar response of the plants against the infection of the fungus M. cannonballus.

Gamma-aminobutyric acid (GABA) is a 4 carbon non-protein amino acid that is involved in several metabolic functions of plants. For instance, it regulates pH and osmotic potential as well as the growth of pollen tubes and more remarkably, it prevents the accumulation of reactive oxygen species when plants are under stress. Hence, the spike in the concentration of this amino acid may have been related to the stress underwent by the plant due to pathogen infection. Similarly, Kinnersley & Turano (2000) documented that the production of GABA is linked to low pH concentrations. Thus, the use of peat moss as a medium, may have contributed to its production.

The existence of higher concentrations of GABA may be indicative of the synthesis of secondary metabolites that might have been used as defensive compounds. Kinnersley & Turano (2000) documented that GABA could be a potential source of carbon to replenish intermediaries in the Krebs cycle that are used to produce secondary metabolites with antimicrobial properties such as phytoalexins, coumesterol and coumarin. Furthermore, its presence has been related to increased tolerance to heat stress in mung beans plants (Priya et al., 2019).

Hydroxy proline was also detected in this study. Generally, the presence of this amino acid is linked to abiotic stress responses related to salinity and drought. Moreover, its presence in cells regulates osmotic potential, protects enzymes as well as membranes. It also captures and retains reactive oxygen species to prevent oxidative damage (Rabisa et al., 2021). Thus, its presence may be associated with a defensive response of the plant. However, it may have not been helpful in countering the infection of the pathogen since it was up regulated in higher quantities in the susceptible genotype. Hence, it could be linked to its susceptibility (Table 1) .

Amino acids such as valine, threonine, methionine, phenylalanine, isoleucine, and leucine are commonly found in melons. Moreover, they are linked to characteristics related to quality. For example, fruit aroma, nutritional value, and health-promoting properties (Singh et al., 2020).

Current knowledge indicates that the catabolism of glycine and serine is energetically expensive. Hence, they usually remain in the plant without any change. On the other hand, glutamine is metabolized to aspartate and glutamate, which is a precursor of GABA (Hildebrandt et al., 2015).

It is notable that some amino acids such as lysine and tryptophan were not present in the roots (Table 1). Lysine is required for the synthesis of L-pipecolate, which is a regulator of inducible plant immunity and tryptophan is a precursor of auxins and secondary metabolites such as phytoalexins, glucosinolates and alkaloids (Hildebrandt et al., 2015; Navarova et al., 2012; Radwanski & Last, 1995).

Conclusions

Herein, it is shown that the production of some amino acids during the interaction of the susceptible and the resistant genotypes of melon plants to vine decline disease with the fungus M. cannonballus is different regarding quantities over time. Interestingly, the existence of higher quantities of gamma-aminobutyric acid in the susceptible genotype TAM-Uvalde 48 and 72 h after inoculating the fungus, M. cannonballus, suggests more penetration of the pathogen in its roots. Furthermore, higher quantities of hydroxy proline were always present in the susceptible genotype TAM-Uvalde after inoculating the fungus, which could be interpreted as an indicator of its susceptibility. Thus, taken together both amino acids may be used as markers of susceptibility to vine decline disease caused by M. cannonballus. Finally, more research is needed to ascertain the role of citrulline.

Supplemental Information

Supplemental Information 1 Concentration of amino acids

Each data point indicates the concentration of amino acids per treatment per repetition

Click here for additional data file.

We want to express our gratitude to Dr. Tom Isakeit, professor and extension specialist of the department of plant pathology and microbiology at Texas A&M University, for his help in conducting this research and providing the pictures of the fungus M. cannonballus.

Additional Information and Declarations

Competing Interests

Author Contributions

Data Availability

The authors declare there are no competing interests.

Sixto Alberto Marquez conceived and designed the experiments, performed the experiments, analyzed the data, prepared figures and/or tables, authored or reviewed drafts of the article, and approved the final draft.

Kevin Crosby conceived and designed the experiments, authored or reviewed drafts of the article, and approved the final draft.

Bhimanagouda Patil conceived and designed the experiments, authored or reviewed drafts of the article, and approved the final draft.

Carlos Avila conceived and designed the experiments, authored or reviewed drafts of the article, and approved the final draft.

Amir M.H. Ibrahim conceived and designed the experiments, authored or reviewed drafts of the article, and approved the final draft.

Herika Pessoa conceived and designed the experiments, performed the experiments, authored or reviewed drafts of the article, and approved the final draft.

Jashbir Singh conceived and designed the experiments, performed the experiments, authored or reviewed drafts of the article, and approved the final draft.

The following information was supplied regarding data availability:

The raw data is available in the Supplemental File.

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
