# Peer review of "Hydroxy proline and gamma-aminobutyric acid: markers of susceptibility to vine decline disease caused by the fungus Monosporascus cannonballus in melons (Cucumis melo L.)"

_PeerJ, doi:10.7717/peerj.14932_

## Round 0.1 · original submission · Major Revisions

Dear Dr. Marquez,

Thank you for your submission to PeerJ. Based on the review reports and my own assessment of your manuscript, I have come to the conclusion that you need to thoroughly revise the manuscript, addressing all the queries raised by the reviewers.

Specifically, you have to consider one suggestion pertaining to combining all the data in tables into a single table for the sake of comparison.

Further, as desired by the reviewers, you have to provide good quality photographs on aspects like pathogenicity, culture, and spore morphology of the pathogen, and also a field photo of vine decline disease to enhance the quality of the manuscript for prospective readers.

Please rewrite the sentences and paragraphs, wherever applicable as per reviewers' suggestions.

The conclusions part in the abstract is missing, and it has to be added.

I hope you will address all the points raised by the reviewers within the stipulated time. Although not a hard deadline please try to submit your revision within the next 35 days.

·

Basic reporting

Clear and unambiguous, professional English used throughout.

Experimental design

Methods described with sufficient detail & information to replicate

Validity of the findings

All underlying data have been provided; they are robust, statistically sound, & controlled

Additional comments

Suitable for publication a few minor corrections.

1. 3 and 47: In title of the manuscript Monosporascus cannonballus should be Italic and
Markers in small letter.
2. 82: write VVD (vine decline disease) throughout the manuscript
3. 83-90: Write the level of resistance and susceptibility of these two genotypes used in the
experiment.
4. 158: What was the basis of selecting a different time for quantitative analysis of amino
acids?
5. 241-242: Hydroxy proline was also detected in this study. Generally, the presence of this
amino acid is linked to stress responses. Which type of stress, either biotic or abiotic?
6. Is there any role of citrulline on susceptibility or resistance of genotypes?

Reviewer 2 ·

Basic reporting

The manuscript entitled “Hydroxy proline and gamma-aminobutyric acid: Markers of susceptibility to vine decline disease caused by the fungus Monosporascus cannonballus in melons (Cucumis melo l.) was need major revision following points:

1.Abstract needs major corrections that were marked in the original MS
2. Many parts of the Introduction sentences are very poorly written and updated with recent literature
3. Material and methods: Give proper protocol for how you isolated the pure culture of the pathogen from roots, Pathogenicity of the pathogen not proved, also provide good photographs of the pathogen (cultural and spore morphology) so that your article quality will be increased. Rewrite some parts in the method of the material as marked in the MS
4. Result and discussion: Here only quantified the different AA compositions in susceptible and resistant genotypes during host-pathogen interaction, but not validated the data with other melon varieties
5. Combine the all table (1 to 4) so that it will be easy for comparison
6. Write the pathogen name always M. cannonballs not MC
7. Provide good photographs of pathogenicity, culture, and spore morphology of pathogen
8. Provide a good field disease photograph so that reader will know how this disease under field condition
9. Discussion is too long, remove the unwanted sentences, discuss only melons
10. Overall my suggestion is to rewrite the entire MS and re-submit to the journal after the correction

Experimental design

.

Validity of the findings

.

Additional comments

.

Annotated reviews are not available for download in order to protect the identity of reviewers who chose to remain anonymous.

·

Basic reporting

It is clear and unambiguous. it is well written manuscript and professional english has been used throughout the manuscript. Literature cited seems to be sufficient and appropriate. Structure of the the manuscript as per the Journal.

Experimental design

Original research and will be helhful to identfy the susceptibility factor of vine decline disease through metabolite profiling.

Validity of the findings

Findings are impactful and meaningful. It is supported by robust data.

Additional comments

Findings of this manuscript will be helpful in distinguishing the resistant and susceptible melon genotypes against vine declining disease.

·

Basic reporting

Marquez et al describe an important study with the potential to increase the knowledge about the host-pathogen interaction while providing insightful information to improve the culture of melons and consequently improve the economic side and reduce the use of chemicals which pollutes the environment.

The suggestion of GABA and hydroxy proline as two markers of susceptibility to VDD is an important finding described in this manuscript, however the authors should revise the results and discussion section in order to show the data in a more exciting manner instead of 4 independent tables. It would also be of interest for the quality of the manuscript if the authors explore more the oscillation of the GABA values detected.

The conclusion would benefit from a short paragraph mentioning who these markers can be further used and implemented for detection of susceptibility to VDD.

Experimental design

The experimental design could include more detail.

1) The inclusion of a figure which could explain the various steps to the reader, for example. It is important to refer the all the information that will allow others to repeat the experiments. For example, the use pf V8 plates, if they are homemade, it would be good to have the composition; if they were bought from commercial venders, then include the reference for it.

2) Perhaps more information about the peat moss and all the components used for the plant growing since these will determine a starting point of conditions like pH, which can have an important role on the plant response and consequently implications of the amino acid composition explored in the study.

3) It is mentioned that roots were sampled, but it is not clear how many plants were used per condition, per time point. It is important to refer the number of replicates and how those replicates were treated during the sample preparation and the data analysis.

Validity of the findings

The results section starts with a sub-section “Qualitative analysis of amino acids” however the description throughout that section is not qualitative as the authors don’t mention any values while referring to the data present on the tables and using only the expression “significantly higher”.

4) The description of the data presented on tables 1-4 should be done in a better way throughout out the results sections “0 hours’, “24 hours” and “48 hours”. The authors should describe their results not only referring to the tables, but also pointing to the values present of those tables. The manuscript would be clear for the reader if the data could be shown in different ways. Perhaps the tables can be supplementary tables, or just one simple table, and the major results shown in individual plots showing the amino acid amount (y) amongst time (x) per condition, for those amino acids which are detected at least in one of the time points, or only for the ones of more interest. For sure, some sort of plot at least for the two amino acids referred as marker candidates, would make the message much clear.

5) The variation of GABA quantification along the time and between conditions needs to be more explored. The authors make a good point explaining the increase of GABA from inoculation to 24h post-inoculation for the RI and SI conditions, however they don’t have an explanation why the concentrations of GABA decrease in the same interval of time for RC and SC conditions. Also, what is the explanation for the GABA values increase from 0 at the time point 48h in the RC and RI, to values much similar to those of SC and SI at 72h?

6) Since the authors suggest GABA and hydroxy proline as markers of susceptibility, it would be good to explore a larger window post inoculation for these two marker candidates. Did the authors looked at other time points post inoculation other than the ones mentioned on the manuscript?

7) How do the authors suggest using these two candidates as markers? What is the time frame to test for the concentrations of GABA and hydroxy proline to be on a safer detection window, especially for GABA which present oscillation on detectability?

8) Is there a way to use these markers in different areas of the plant? Assuming the use of these markers to detect susceptibility to VDD, it would be good if the plants can be sampled and tested in a less invasive way. Do the authors have information about these two markers in other areas other than roots for the same time frame?

9) It come to my attention a lot of values that are exactly the same for different samples. It was very surprising for me that even if two samples have a similar value, that the value is exactly the same. Can the authors explain why this happens? I assumed on table 1 that the values would be the same for both samples of each genotype as this was pre-inoculation, although this should still have some variability and I would expect the values to be closer between two samples of the same genotype, but not the same exact value.

10) In the tables, it would be important to make. Distinction between 0 and n.d if any? Also, the authors need to be more descriptive about the meaning of a, b and c on the table legend description.

Additional comments

Line 47 - Monosporascus Cannonballus should be Monosporascus cannonballus<i>

Line 56 until the end of the manuscript - Monosporascus cannonballus should be M. cannonballus

Line 76 – Ref Crosby, 2000 is not listed of reference list

Line 104 - 1.22 107 should be 1.22x107

Line 150 – Identification should be identification

Line 313 – Ref Bush 1999 is not cited in the manuscript

---

## Round 0.2 · accepted · Accept

Based on review report and my own assessment, the article now appears in good shape. The current version is ready for publication.

The Section Editor noted:

> Centrifugation force should be given in 'g,' not in 'rpm.'

Please correct this in the proof stage.

·

Basic reporting

The authors have carefully made changes and adding information according the reviewers comments on previous version of the manuscript. The union of the various tables in one single table was the major request and was fully completed, making reading and the comparison easier for the reader.
The addition of new figures to illustrate the what was described in the manuscript was also completed and certainly made the improvement.

Experimental design

N/A

Validity of the findings

N/A

Additional comments

N/A